# Performance of Polyethylene Vapor Barrier Systems in Temperate Climates

**Torben Valdbjørn Rasmussen** [1,*], **Tessa Kvist Hansen** [1], **Yvonne Shashoua** [2], **Lisbeth M. Ottosen** [3], **Louise Green Pedersen** [3], **Jens Kromann Nielsen** [4] and **Frederik R. Steenstrup** [4]

1. Department of Civil Engineering and Construction Management, BUILD, Aalborg University, 2450 Copenhagen, Denmark
2. Environmental Archaeology and Material Research, National Museum of Denmark, 2800 Kongens Lyngby, Denmark
3. Department of Civil Engineering, Technical University of Denmark, 2800 Kongens Lyngby, Denmark
4. Department of Plastics and Packaging Technology, Danish Technical Institute, 2630 Taastrup, Denmark
* Correspondence: tvr@build.aau.dk; Tel.: +45-2360-5697

**Abstract:** The performance of nine different vapor barrier systems comprising polyethylene (PE) membranes were assessed. The vapor barrier systems comprised membranes of virgin PE, 100% new PE, regenerated PE and multilayered virgin and regenerated PE. Membranes were joined either with tape suited to the individual system or an adhesive base on butyl rubber. The vapor barrier systems were evaluated and compared using standard laboratory tests. Chemical analytical techniques and physicomechanical tests were used. Mechanical properties were assessed using laboratory tests recommended by the harmonized standard EN 1385. Chemical analyses followed standard laboratory protocols performed with specialized equipment and visual examination. Chemical and mechanical properties were determined before and after exposure to an aging regime comprising 168 days at 70 °C in total. The chemical stability of the plastic present in each membrane was further evaluated after an additional exposure to an aging regime comprising 50 days followed by another 30 days at 70 °C. Additional aging indicated chemical changes in the membrane material with time. However, it was not possible to distinguish between aging properties for membranes containing virgin PE, 100% new PE, regenerated PE or multilayered virgin and regenerated PE.

**Keywords:** laboratory tests; vapor barrier systems; polyethylene; aging; virgin PE; new PE; regenerated PE; membranes; joints

## 1. Introduction

Vapor barrier systems are used in temperate and cold climates to ensure the required airtightness and to reduce diffusion of water vapor into the building envelope [1,2]. The temperate climate zones are located in the regions of the earth between the tropic regions and the polar regions. Temperate climates are characterized by relatively moderate mean annual temperatures, with average monthly temperatures above 10 °C in their warmest months and above −3 °C in their colder months [3]. Most regions with a temperate climate present four seasons, and temperatures can change greatly between summer and winter [4]. In these climates, airtightness is also required to reduce energy consumption and is specified as a requirement in some building regulations, e.g., [5]. Reducing diffusion secures against high levels of moisture within the thermal insulated building envelope [6]. Independently of whether an air barrier or a vapor barrier is needed, the tightness of the joints between its components is crucial [2]. Furthermore, the joints must stay tight throughout the service life of the membrane. In this case, joints are defined as the seam between two lengths of membrane, and joints between membrane and other building parts. Tapes or adhesives can be used to seal joints. Manufacturers sell membranes and sealants as system solutions, preferably including collars for any pipes that may penetrate the membrane. Using a

system solution is a way to ensure that membranes and sealants are compatible and a system solution is usually required by manufactures to guarantee product performance.

Polyethylene (PE) membranes are frequently used as vapor barriers that, together with a specified tape, perform as system solutions. In the past, it was common for PE membranes to be made from pure (virgin or 100% new) PE, but today, an increasing number of membranes on the market contain regenerated, or a mixture of regenerated and pure polyethylene. As part of the increased interest in sustainability in construction [7–10], it is expected that demand will increase for fully or partially-regenerated membranes.

Virgin PE is defined as a material that has been fabricated only once. Whereas 100% new PE is defined as a material that has not been used after fabrication as well as offcuts and residual material from the production of virgin PE. This 100% new PE contains a significant proportion of the additives from the production of virgin PE. The quantity and the types of additives in 100% new PE varies. Regenerated PE is fabricated from collected, post-use plastic, also called recycled plastic. The plastic is sorted, cleaned, and washed before it is melted and supplemented with additives. A combination of pure and regenerated PE is used to fabricate a layered membrane. The base layer comprises regenerated PE and thinner layers of 100% new PE or virgin PE are added to one or both sides of the base layer.

Vapor barrier systems in temperate and cold climates are mounted on the warm side of the thermal insulated exterior building envelope [11]; see Figure 1. Knowledge and experience of vapor barrier systems comprising membranes of pure PE (100% new PE and virgin PE) has been acquired through years of use. However, knowledge and documentation of the performance, particular in the long-term, of newer vapor barriers containing regenerated PE membranes are lacking [12].

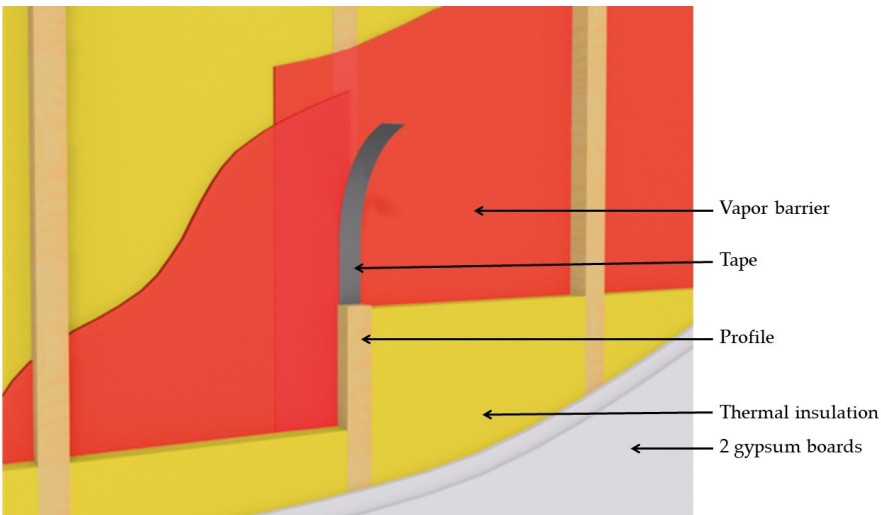

**Figure 1.** Exterior timber stud wall, where the vapor barrier system (shown in red) with a joint between two lengths of membrane sealant with tape (shown in grey), is mounted on the warm side of the thermally insulated exterior building envelope.

This paper assesses the long-term performances of vapor barrier systems comprising PE membranes joined with the appropriate tape or butyl adhesive. Furthermore, the paper identifies, evaluates and compares the compositions, durability and long-term performances of vapor barrier membranes comprising virgin PE, 100% new PE, regenerated PE and a single membrane comprising PE layered with virgin and regenerated materials. Commercially available vapor barrier systems comprising nine different membranes were selected for examination and purchased. Membranes were assessed both before and after accelerated thermal aging using chemical analyses and physicomechanical tests to determine the material properties and stabilities of the membranes as well as the strength of joints.

## 2. Materials and Methods

Tests were performed to determine the material properties of vapor barrier systems and the stabilities of the membranes. Samples from vapor barrier systems were assessed using six types of chemical analyses and 13 physicomechanical tests. Samples comprised both membranes and joined membranes using adhesives as tape and butyl, respectively. All test methods were based on standardised methods described by the standard [13] and in ISO or DIN standards. The exceptions were the chemical analyses that followed standard laboratory protocols performed with specialized equipment and visual examination. Most of the tests were performed both before and after aging.

### 2.1. Materials

The tested vapor barrier systems were selected from those most commonly used by contractors in Denmark. Nine different vapor barrier systems were selected for the study comprising nine different PE membranes. PE membranes comprised two fabricated from virgin PE, two from 100% new PE and five from regenerated PE. One of the five membranes of regenerated PE, according to its manufacturer, consisted of a base layer of regenerated PE between two layers of pure PE. One roll of each membrane and associated tape was purchased from retail building material businesses located randomly in Denmark to maintain independency from the suppliers. An additional five rolls of each of the membranes containing regenerated PE were purchased for visual examination. Each roll contained between 250 and 500 m$^2$ of membrane. The properties of the membranes (identified by ID numbers) provided by the manufacturers are shown in Table 1. Membranes comprising virgin PE were identified as T7482-1 and T7482-2. Membranes comprising 100% new PE were numbered as T7482-3 and T7482-4. Membranes comprising regenerated PE were known as T7482-5, T7482-6, T7482-7, and T7482-9. One membrane, denoted as T7482-8 contained three layers of PE, comprising one base layer of regenerated PE sandwiched between two layers of virgin PE. ID numbers assigned to membranes are also used for presenting results for adhesive strength of taped joints for vapour barrier systems and adhesive strength of joints using butyl.

**Table 1.** Properties of membranes evaluated in the study as supplied by manufacturers.

| ID Number | Service Life * (Years) | Membrane Material | Thickness [mm] |
|---|---|---|---|
| T7482-1 | >30 | Virgin PE | 0.20 |
| T7482-2 | 50 | Virgin PE | 0.12 |
| T7482-3 | 50 | 100% new PE | 0.20 |
| T7482-4 | >15 | 100% new PE | 0.20 |
| T7482-5 | >20 | Regenerated PE | 0.15 |
| T7482-6 | >15 | Regenerated PE | 0.20 |
| T7482-7 | >20 | Regenerated PE | 0.20 |
| T7482-8 | >30 | Regenerated PE and pure PE | 0.20 |
| T7482-9 | undocumented | Regenerated PE | 0.20 |

* Service life specified in data sheets provided by the individual manufactures.

### 2.2. Aging

Accelerated thermal aging was used to predict the long-term performance of membranes within a convenient period of time. All evaluation tests were performed both on unaged and aged samples, except for the chemical tests comprising Beilstein test, X-ray fluorescence spectroscopy (XRF) and the Loss on Ignition test which were only conducted on unaged samples and not expected to change with time, see Table 2.

**Table 2.** Chemical analyses of membranes.

| Analysis Type | Performed on Unaged Samples | Performed on Aged Samples |
|---|:---:|:---:|
| Attenuated Total Reflection—Fourier Transform Infrared (ATR-FTIR) spectroscopy | ✓ | ✓ |
| Acid-Detection (A-D) indicator strips | ✓ | ✓ |
| Beilstein test | ✓ | |
| X-ray fluorescence spectroscopy (XRF) | ✓ | |
| Scanning electron microscopy (SEM) | ✓ | ✓ |
| Loss on Ignition | ✓ | |

Aging was performed by placing the samples for 84 days in a climate chamber at 70 ± 2 °C and 90 ± 2% relative humidity (RH), followed by 84 days at 70 °C and RH lower than 10 ± 2% RH to examine the effect of change in RH. In addition, samples of membranes were exposed to an extended aging period comprising 80 days in a climate chamber at 70 °C and RH lower than 10 ± 2% to further examine long term performance of membranes.

When aging plastic, temperature is often used as the accelerating parameter. The higher the temperature, the faster the evaporation of the volatile substances in the material, and the faster changes occur in the molecular structure and thus the mechanical properties of the material. The relationship between thermal aging and the choice of temperature can generally be described by the Arrhenius' velocity expression [14], relating the rate of aging, k, the time, t, and the temperature, T. The relationship may be described by the expression:

$$k = A \times \exp\left(-E/(RT)\right), \tag{1}$$

where:

E = the activation energy for the primary degradation reaction
A = the pre-exponential factor
R = the universal gas constant [J/(mol·K)]
T = temperature [K].

From the expression, it is clear that the speed of the aging or degradation reaction doubles each time the temperature is increased by 10 °C. Holmström and colleagues have investigated whether this also applies to LDPE, and the work is referenced in Verksnorm, 2000, published by Sveriges Plastförbund [15].

Assuming that a vapor barrier when used in the finished building has an average temperature of 10 °C, an aging temperature of 70 °C will correspond to: $2^{(60/10)} = 64$ times faster aging. Additionally, 100 days aging at 70 °C corresponds to 6400 days, i.e., natural aging of approximately 18 years at 10 °C. Assuming that the vapor barrier when used in the finished building has an average temperature of 20 °C, 168 days aging at 70 °C corresponds to natural aging of approximately 15 years at 20 °C. Moreover, 218 (168 + 50) days aging at 70 °C corresponds to approximately 19 years of real time, calculated on the basis that the service temperature of the membranes and assembled systems is 20 °C and 248 (168 + 50 + 30) days aging at 70 °C corresponds to approximately 22 years of real time.

### 2.3. Chemical Analyses

Five samples of each membrane were analysed to determine their chemical structures both before and after aging. The analyses performed are shown in Table 2.

### 2.3.1. Attenuated Total Reflection—Fourier Transform Infrared (ATR-FTIR)

ATR-FTIR spectroscopy was used to identify the chemical structure of the polymer and additives of each membrane and to evaluate their chemical stabilities. To investigate whether samples showed signs of chemical degradation with time, ATR-FTIR spectra of aged samples were compared with those of unaged samples [16]. Samples with the size 50 × 50 mm were used.

### 2.3.2. Acid-Detection (A-D)

A-D indicator strips were used to show the release of volatile organic acids from membranes which indicated a potential reduction in service life of the materials. A-D strips are paper-based indicators that use the pH-sensitive indicator bromocresol green to detect the presence of acids evolved by all materials, including plastics. Their advantage over universal pH indicator papers is that they are used dry and do not require water. When A-D strips are placed in a closed container with the specimen, the strips change colour from blue at pH 5.4 to green (slightly acidic) and then to yellow at pH 3.8 (highly acidic) if acid gases are released. The colour changes within 24 h [16,17]. Samples with the size, $50 \times 50$ mm were used.

### 2.3.3. Beilstein Test

The Beilstein test [18] was used to identify the presence of the plastic type polyvinyl chloride, PVC in samples. PVC is chemically unstable, can emit toxic plasticizers and hydrogen chloride and is therefore undesirable in building materials designed for long term use [16].

For the test, a powerful gas flame from a gas burner, for example, and an approximately 30 cm long copper wire was used. The tip of the copper wire was heated until it glowed, and all impurities were burned away. Sheets of specimens were touched with the tip of the copper wire. The copper wire with plastic was then returned to the flame. If the flame burned green, PVC was detected. If not, then PVC was not present in the sample.

### 2.3.4. X-ray Fluorescence Spectroscopy (XRF)

XRF was used to determine the elemental composition of samples. XRF is a non-destructive, surface measurement technique that can in principle detect and measure the content of all elements, however with a reduced sensitivity for the lightest elements [19]. A Bruker Tracer III-V, that detect elements with a higher atomic number than magnesium was used [16]. XRF spectrometer head was used on sheet-like samples of $1 \times 1$ m.

### 2.3.5. Scanning Electron Microscopy (SEM)

SEM was used to examine the surfaces of plastic sheets ($15 \times 15$ mm) non-destructively. SEM uses electromagnetic radiation to obtain a greater resolution than can be obtained by ordinary light microscopy, and one can thus observe objects down to the nanometer scale [16]. Instead of glass lenses, which are used in light microscopy, SEM uses electric and magnetic lenses, which focus the electrons in a similar way to that used by optical lenses to focus light.

### 2.3.6. Loss on Ignition

Loss on Ignition was used to quantify the content of organic material in samples. Samples were annealed at a temperature of 550 °C and the weight loss was measured. Polyethylene is fully decomposed at this temperature, whereas inorganic additives and mineral impurities will only be decomposed by much higher temperatures. PE is annealed to a weight percent of 0% at 480 °C [20]. Samples of 2–2.5 g were cut from each membrane, pre-dried at 105 °C and then annealed at 550 °C in a muffle furnace. The mass of the pre-dried samples ($m_0$) and the mass after annealing ($m_g$) were weighed to 4 decimal places. The loss on ignition is defined as weight loss on annealing and calculated as $((m_0 - m_g)/m_0) \times 100$ [16].

### 2.4. Physicomechanical Tests

Five samples of each membrane (the water vapor diffusion resistance test was conducted on two samples of each membrane, and adhesive strength conducted on seven samples) were analysed, using 13 physicomechanical tests. The tests were performed both on unaged and aged samples. Bending force was only conducted on unaged membranes. The tests performed, [12–28] are shown in Table 3.

**Table 3.** Physicomechanical tests performed on membranes.

| Test Type | Test Standard | Performed on Unaged Samples | Performed on Aged Samples |
|---|---|---|---|
| Tensile strength, both longitudinal and transverse to the production direction fracture elongation and Young's modulus | DS/EN ISO 527-3 | ✓ | ✓ |
| Tear strength both longitudinal and transverse to the production direction | DS/EN 12310-1 | ✓ | ✓ |
| Resistance to impact | DS/EN 12691 | ✓ | ✓ |
| Mass per unit area | DS/EN 1849-2 | ✓ | |
| Thickness | DS/EN 1849-2 | ✓ | |
| Water vapor diffusion resistance | DS/EN 1931 | ✓ | ✓ |
| Surface tension | ASTM D2578 | ✓ | ✓ |
| Adhesive strength | DS/EN 12316-2 | ✓ | ✓ |
| Bending force | DS/EN ISO 178 | ✓ | |

### 2.4.1. Tensile Strength, Fracture Elongation and Young's Modulus

The tensile strength was determined as the highest measured stress during the test. Elongation at break was determined as the strain at break. Young's modulus was determined at 0.5–1.0% elongation, although it is not part of the standard used [21]. Samples are shaped like meat bones, according to the specification described in [21], Type 5 in the specification. The size of the meat bones-shaped specimens were as follows: the width of the narrow part of the bone was $6 \pm 0.4$ mm; the length of the narrow part of the bone was $33 \pm 2$ mm; the width of the ends of the bone was $25 \pm 1$ mm; the overall length of the bone was a minimum of 115mm, and the initial distance between the grips was $80 \pm 5$ mm. Tests were performed on a tensile testing machine of the type: Shimadzu machine model AG-X, class 0.5 load cell 500 N, see Figure 2. To measure the strain, a click-on extensometer from Shimadzu, class 1 was used. To measure the thickness of the sample, a micrometer was use of the type: Mitutoyo Thickness gauge. Tests were performed at $23 \pm 1$ °C and $50 \pm 2$% RH. The towing speed was 100 mm/min.

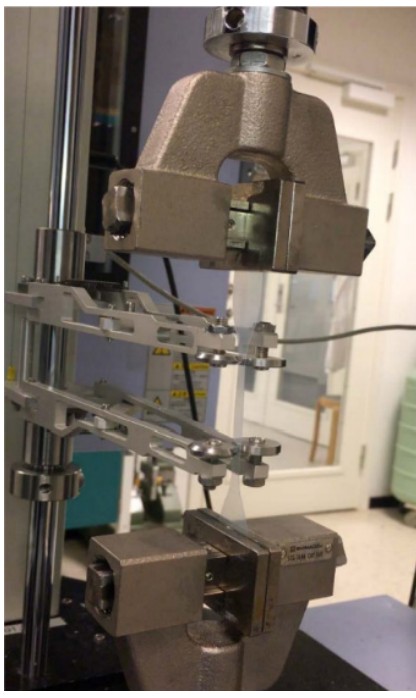

**Figure 2.** Sample mounted in the tensile testing machine.

### 2.4.2. Tear Strength

The tear strength was determined as the highest measured strength during the tear strength test. Tests were performed on a tensile testing machine of the type: Instron tensile testing machine equipped with a 5 kN load cell. Samples shaped like sheets, according to the specification described in [22] of 100 × 100　mm in size were mounted in the tensile tester in a sample fixture, as shown in Figure 3, and perforated with a mandrel for this purpose. The fixture is also shown in [22]. The mandrel was pulled through the sample and the traction force was measured continuously during the experiment. Tests were performed at 23 ± 1 °C at 50 ± 2% RH. The towing speed was 100　mm/min.

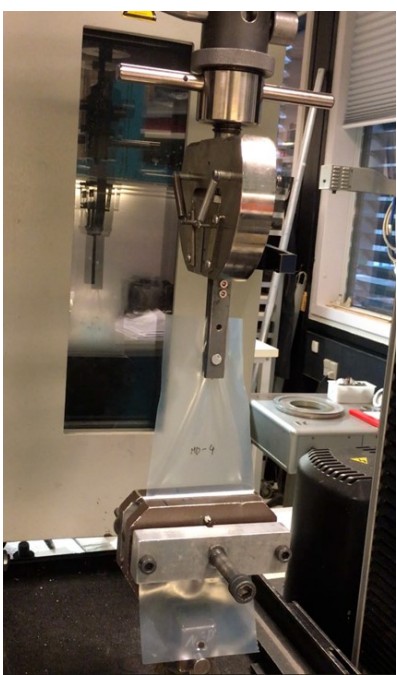

**Figure 3.** Tensile testing in a sample fixture perforated with a mandrel arranged for the purpose.

### 2.4.3. Resistance to Impact

Resistance to impact was determined by measuring the impact at which the membrane was perforated. The resistance to impact was expressed as the maximum height from which a weighted ball can be released without penetrating the specimen. The drop weight, shown in Figure 4 weighs 500 g ± 5 g and the diameter of the ball at the bottom of the drop head was 12.7　mm. The drop weight falls from up to 2 m above the point where the drop weight hits. The distance between the holes in which the split could be placed varies with the height: between 200 and 500　mm height there was 50　mm between the holes, between 500 and 1000　mm there was 100　mm between the holes and for heights above 1000　mm there was 250　mm between the holes. The test specimen was placed on a base consisting of a 3　mm thick aluminium plate with an area of 300　mm × 300　mm. The aluminium plate was moved 25　mm between each measurement. The drop weight sets a mark in the plate; therefore, it was important that the drop weight on the next attempt hits a flat surface without a mark. Below the aluminium plate, a cast tile with a steel plate on top in the same dimensions as the aluminium plate, was placed. Samples shaped like sheets of 100 × 100　mm in size were used. Samples were assessed visually to identify whether a hole occurred. If holes were not visible a vacuum was applied. Resistance to impact was quantified as the height at which a minimum of 4 out of 5 samples passed the test height.

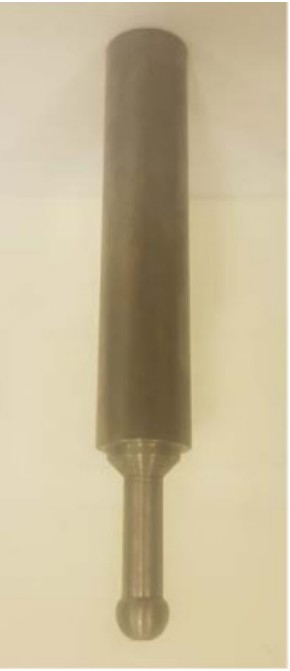

**Figure 4.** Drop weight and ball weighs 500 g $\pm$ 5 g and has a diameter of 12.7 mm.

2.4.4. Mass per Unit Area and Thickness

Samples were taken from the individual membranes according to [29]. Samples in sheet form, $100 \times 100$ mm in size with areas $10{,}000 \pm 100$ mm$^2$ were weighed before aging conditioned at 27 °C and 56.3% RH (basis weight). The thickness was measured mechanically with a dial gauge from Sylvac shown in Figure 5. The thickness was indicated with a precision of 0.01 mm. The flat surface measured had a diameter of 10 mm $\pm$ 0.05 mm under a pressure of 20 kPa $\pm$ 10 kPa.

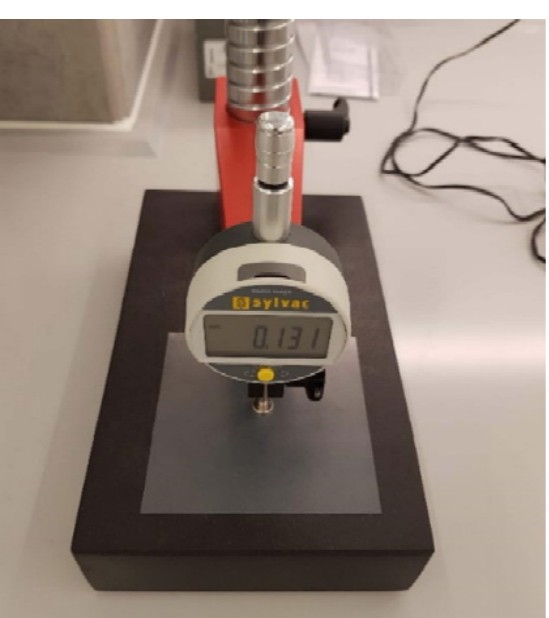

**Figure 5.** Mechanical thickness measurements of a sample using a dial gauge from Sylvac.

The samples were weighed on a scale to the nearest 0.01 g. Results were given in g/m$^2$.

### 2.4.5. Water Vapor Diffusion Resistance

Water vapor diffusion resistance was determined using a cup sealed with a lid consisting of a test membrane, see Figure 6. A saturated saline solution ($KNO_3$) inside the cup ensures a constant relative humidity of 93% ($RF_1$). The cups were placed in a climate chamber at $23 \pm 1\,°C$ and $50 \pm 2\%$ RH ($RF_2$). The lid area was $0.00785\ m^2$. The lid area was identical to the size of samples. The water vapor flux was found by recording the weight change over time. As the moisture flow stabilizes, a straight line is recorded. The slope of the curve was determined describing a constant moisture flow.

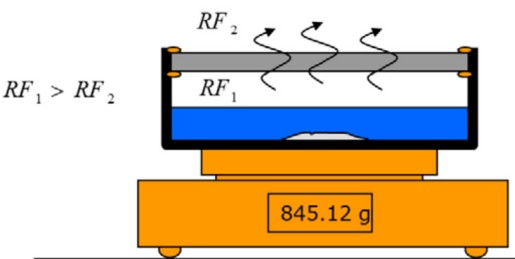

**Figure 6.** A cup closed with a lid consisting of membrane placed on a weight.

The flux through a membrane at a given time can be determined from:

$$q_d = (m_2 - m_1)/(A\ (t_2 - t_1)), \tag{2}$$

where,

$m_i$ is the mass of the cup at a given time, $t_i$, i = 1, 2, [kg]
$t_i$ is the time [s]
A is the area for the flow [$m^2$]

Water vapor diffusion resistance (Z) [GPa s $m^2$/kg] is a material property that describes a materials resistance to vapor diffusion:

$$Z = (p_1 - p_2)/q, \tag{3}$$

where, $p_1$ and $p_2$ are the partial pressures inside and outside the cup respectively. These partial pressures are based on knowledge of RH and temperature during the experiment. The temperature was $23 \pm 1\,°C$. RH1 and RH2 were 93% and 50%, respectively. At $23\,°C$ the saturation pressure is $p_s = 2810$ Pa [30]. Partial pressure $p = p_s \times$ RH, i.e., $p_1 = 2613$ Pa and $p_2 = 1405$ Pa.

### 2.4.6. Surface Tension

Surface tension was determined using the Accu Dyne TestTM pen. A set of pens with different liquids was used, see Figure 7. The method is based on [26] Standard Test Method for Wetting Tension of Polyethylene and Polypropylene (PP) Films, where the surface tensions of PE and PP membranes are determined by measuring contact angles of droplets on the membrane with the same liquids. The Accu Dyne TestTM pens contain the same liquids, but the assessment of the surface tension has been simplified. The result of the test is a "duvet level", which can be used to compare membranes. Information about the pens can be found on the website https://www.accudynetest.com/pentest.html (accessed on 20 October 2022).Three parallel lines are drawn lightly on a piece of membrane, and development of the line within the first three seconds is assessed. If the area below the line remains damp and the line is intact for more than 3 s, the dyne level is too low and the pen with a higher dyne level tested. If, on the other hand, the line separates or becomes thin within one second, the dyne level is too high and a pen with a lower dyne level. The right dyne-level is found when the line holds its shape for one to three seconds before changing form. Membranes were stored for 24 h, and the test was performed at $24-25\,°C$ and $35-38\%$ RH. Sheets of $150 \times 150$  mm in size were used as samples.

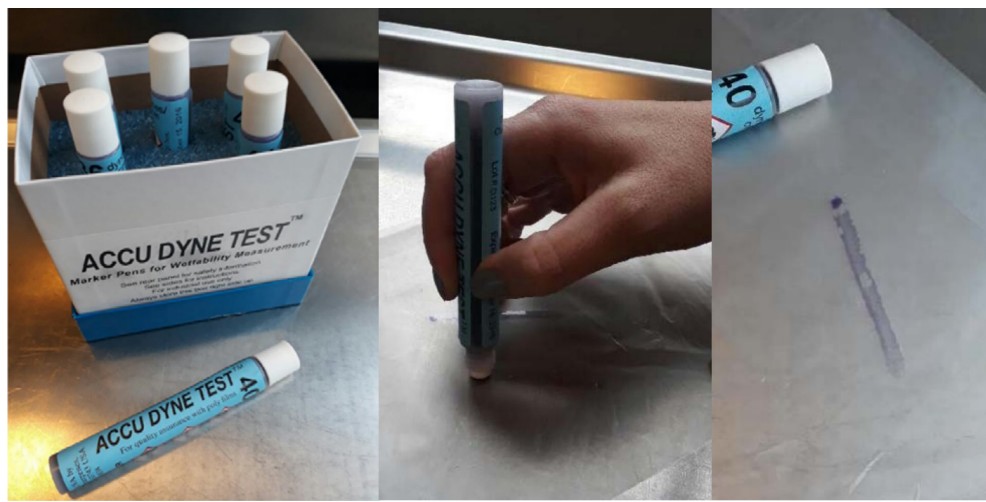

**Figure 7.** Accu Dyne TestTM pen.

### 2.4.7. Adhesive Strength

Adhesive strength of taped joints of membranes and joints adhered with butyl adhesive were measured as described in [27]—Part 2: Plastic and rubber membranes for roofing. Sampling was performed according to [29]. Samples with the dimension $50 \times 100$ mm to the joint, see Figure 8 was formed. Joints were pressed with a steel roll with the following specifications: outer diameter (including rubber bandage): $(85 \pm 2.5)$ mm, button roller weight: $(2000 \pm 10)$ g, width: $(50 \pm 1.5)$ mm, hardness of the rubber band: $(80 \pm 5$, Shore A). Samples tested before aging were conditioned at 24 °C at 54% RH.

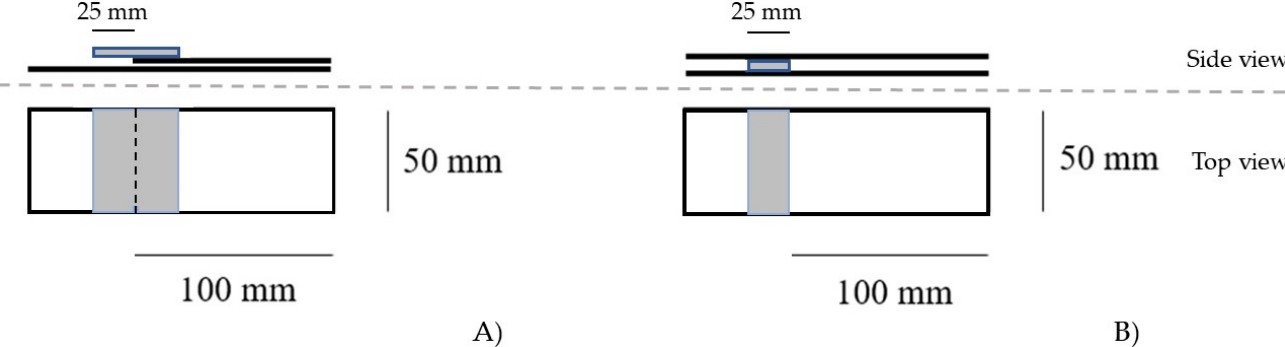

**Figure 8.** (**A**) assembly with single-sided adhesive tape, (**B**) butyl.

The adhesive strength was measured using an Instron test machine, see Figure 9. The adhesive strength was measured at a constant grip operating speed of 100 mm $\pm$ 10 mm/min. The results were given in N.

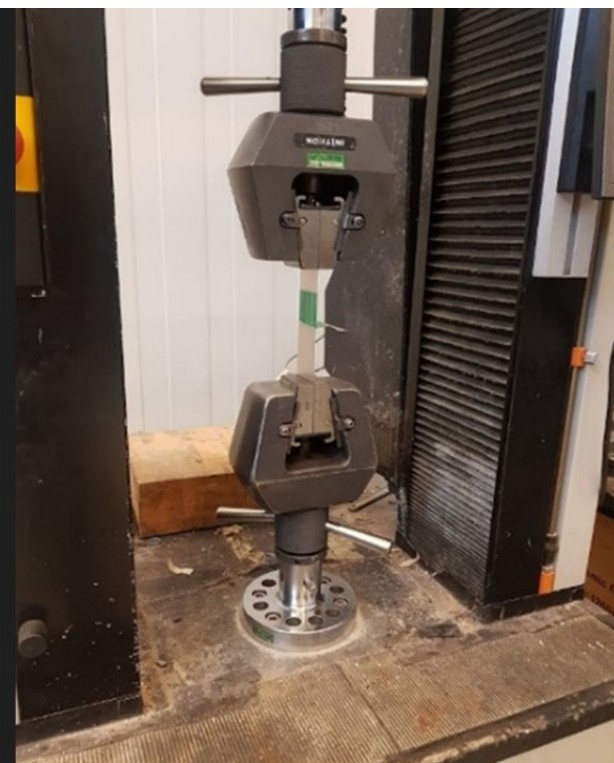

**Figure 9.** Instron test machine for measuring the adhesive strength.

2.4.8. Bending Force

Bending force was determined as described in [28]. The bending force of membranes was compared using an index. The index determines how easily one membrane compared to another can be handled on site, Figure 10. The bending force depends on the bending force, Young's modulus, thickness, width and length of the sample. For the calculation of the bending force, assumptions have been made with regard to d, L and w for comparison purposes. The assumptions were; d = 1 mm, L = 10 mm and w = 1 m. Samples described in the standard [28] were used.

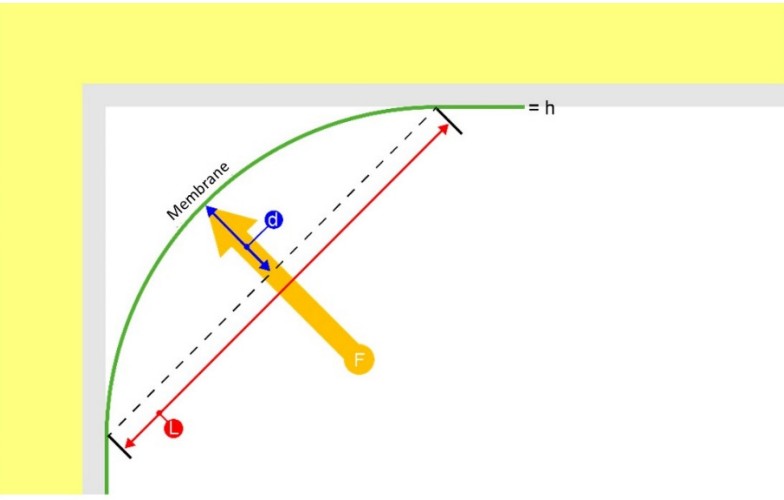

**Figure 10.** Bending force for a membrane.

The bending force is defined as:

$$F = E\,h^3\,((4\,w\,d)/L^3), \tag{4}$$

where,

      F: Bending force [N]
      E: Young's modulus for membrane [MPa]
      h: thickness of membrane [m]
      d: deflection [m]
      L: Length between support points [m]
      w: width of membrane [m].

### 2.5. Visual Examination

Five complete rolls of each membrane containing regenerated PE were examined for cracks, larger fragments and weak areas. Rolls were opened to their full width and length, so they could be fully examined in a single session. The membranes were examined visually in detail, piece by piece.

Cracks, large fragments and weak areas in the individual membrane were cut out and assembled for further classification. The visual examination of membranes thus served to check and compare the sorting and purification of all membranes comprising regenerated PE and its combination with pure PE, had been conducted to an acceptable degree under production.

Detailed information on the manufacturing process was not disclosed by the suppliers, but sorting, purifying, washing, and addition of stabilizers were believed crucial in production of membranes comprising regenerated PE. Membranes of regenerated PE are extruded from molten granules to which stabilizers are added. Granules have been pre-sorted, cleaned and washed. Membranes are extruded and folded before being rolled around a cardboard tube in lengths of typically 25 or 50 m. Rolls were 1 m wide, and the membranes folded in the middle. During visual examination, membranes were unfolded and pulled over a table, 2 m at a time. When the membrane was on the table, the fold-line was examined first followed by the flat surfaces of the membrane itself.

## 3. Results

### 3.1. Chemical Analyses

#### 3.1.1. ATR-FTIR Spectroscopy

ATR-FTIR spectra of samples of unaged, pure PE, virgin PE (T7482-1 and T7482-2) and 100% new PE (T7482-3 and T7482-4), (Figure 11 (left)) before aging were chemically identical to those of reference samples low-density PE (LDPE) run at the same time. All the pure PE samples exhibited characteristic peaks between 3000 and 2840 cm$^{-1}$, attributed to asymmetric and symmetrical stretching of $CH_2$ groups, peaks at 1469 cm$^{-1}$ due to deformation in $CH_2$ groups and at 718 cm$^{-1}$ from rocking vibration of $CH_2$ groups [31]. No polymer types other than LDPE were identified in the pure PE samples.

ATR-FTIR spectra of samples containing regenerated PE (T7482-5, T7482-6, T7482-7, T7482-8, and T7482-9), (Figure 11 (right)) matched those of reference samples of LDPE. The difference between ATR-FTIR spectra of samples of pure PE and samples containing regenerated PE was that samples containing regenerated PE showed additional small peaks between 1500 and 650 cm$^{-1}$ attributed to inorganic additives. No polymers other than LDPE were identified in the samples. All samples containing regenerated PE contained calcium carbonate, but T7482-9 contained measurably higher concentration than the other samples containing regenerated PE. The higher concentration of calcium carbonate in samples of T7482-9 is seen as strong and broad peaks between 1450 and 650cm$^{-1}$.

After exposure of membranes to additional aging for 50 days (in total 218 days) at $70 \pm 1$ °C and RH lower than 10%, changes were detected in at least one specimen of the membranes denoted T7482-3, T7482-6, T7482-8 and T7482-9. ATR-FTIR spectra of unaged samples of membranes contained only C-H bonds, while the development of peaks in the range 1700–1750 cm$^{-1}$ indicated the formation of carbonyl groups (C=O) in additional aged PE. Carbonyl bonds indicated the development of oxidation products of ketones, acids and esters [32] and therefore were a sign of chemical instability in the polymer (Figure 12).

The formation of carbonyl groups is often observed as yellowing in polyethylene. Visual observations of samples of membrane T7482-3, T7482-6, T7482-8 and T7482-9 are shown in Figure 13.

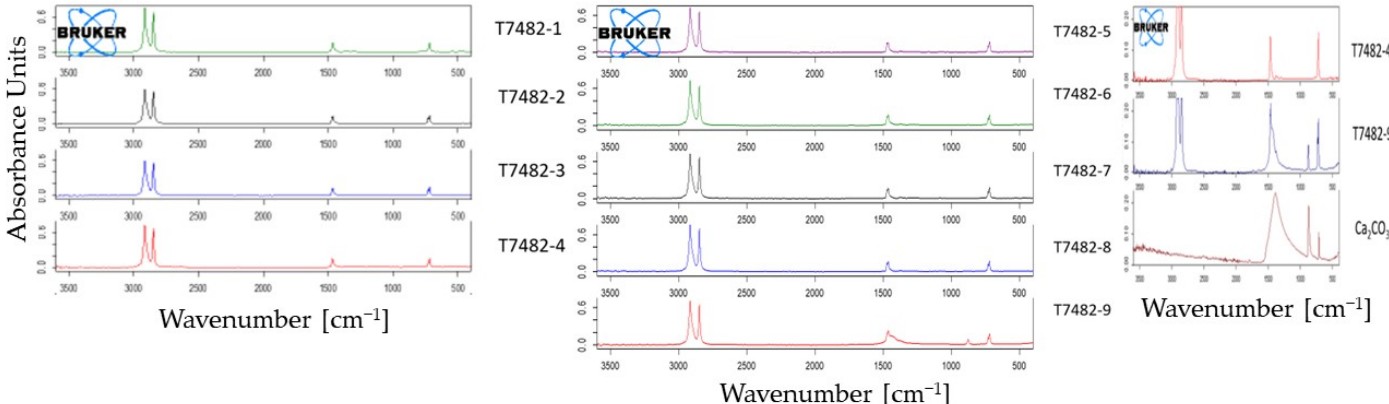

**Figure 11.** ATR-FTIR spectra of membranes of pure PE (**left**) and regenerated PE (**middle**). Spectra of a membrane of pure PE (**left**) compared with regenerated (**middle**) are overlayed with a spectrum of reference calcium carbonate to identify the additional peaks attributed to the inorganic filler material (**right**).

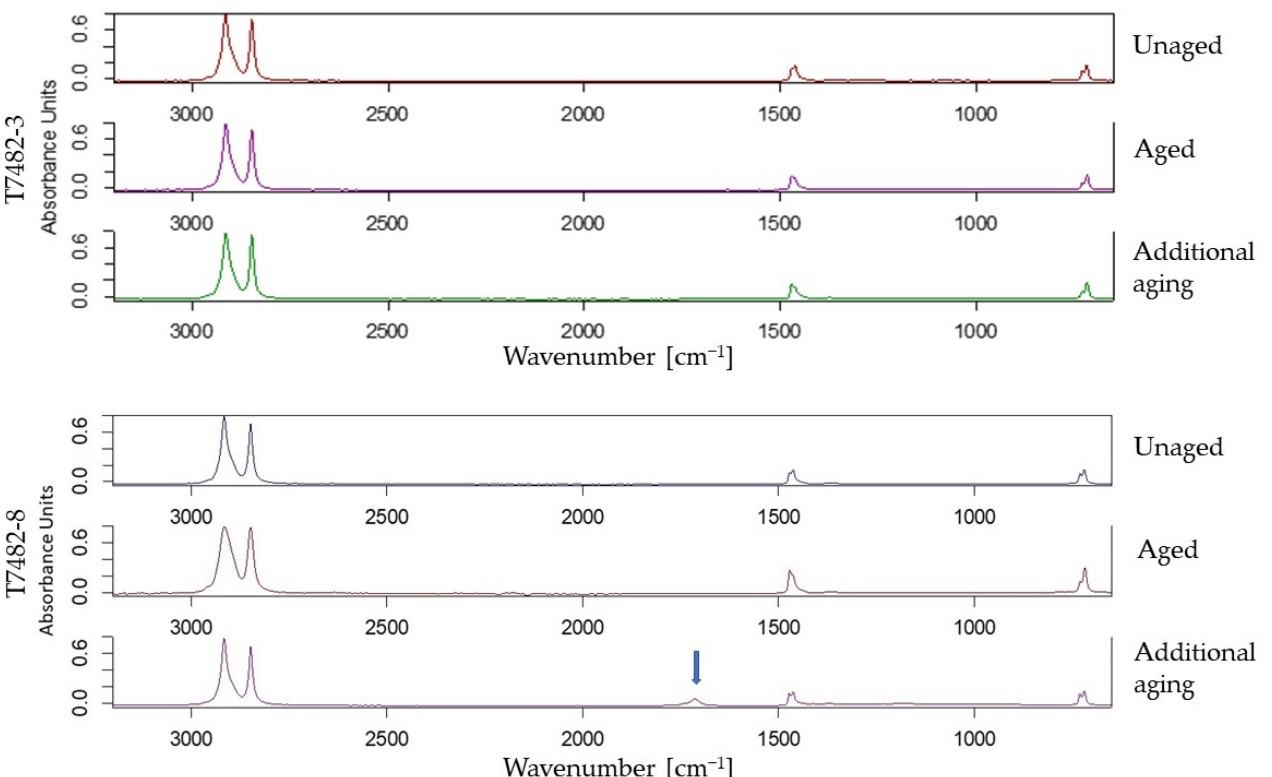

**Figure 12.** In the upper figure, ATR-FTIR spectra of T7482-3 shows no significant chemical changes for unaged samples (**top**), after aging (**middle**) or additional aging (**bottom**). By comparison, in the lower figure, ATR-FTIR spectra of T7482-8 shows no significant chemical changes for unaged samples (**top**), nor after aging (**middle**), but the distribution and formation of new peaks centred around 1720 cm$^{-1}$ due to oxidation, is evident after additional aging for 50 days (**bottom**).

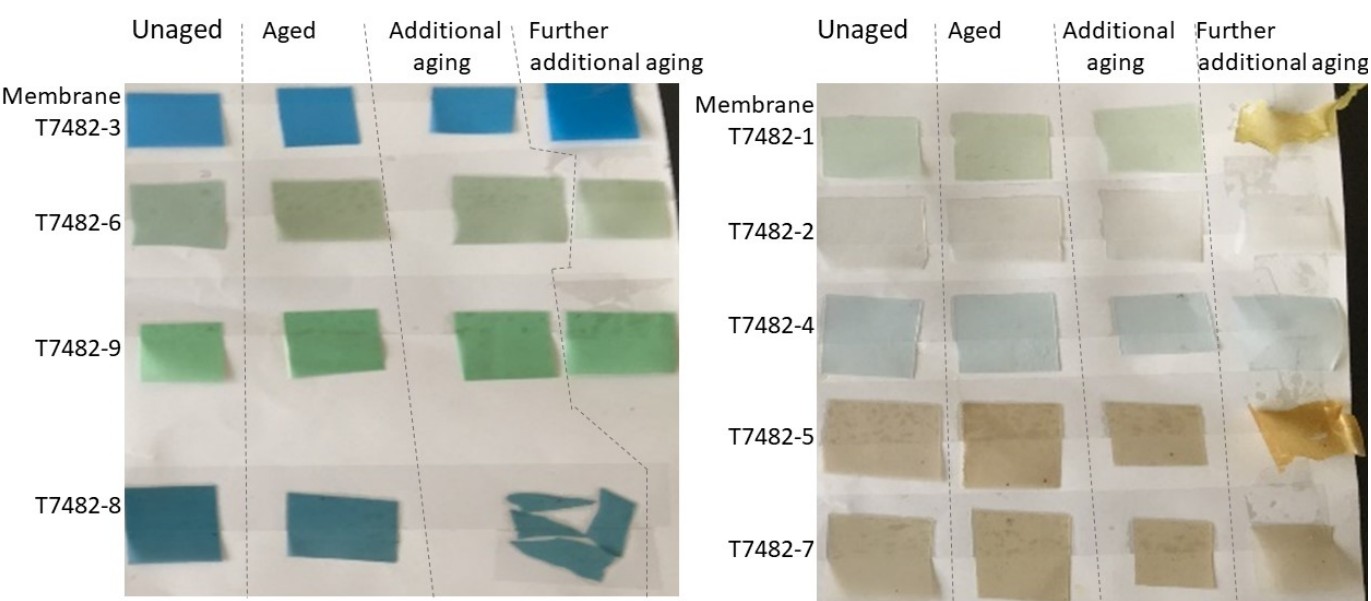

**Figure 13.** Membranes that showed measurable changes in chemical stability. Membrane T7482-8 showed significant chemical changes after additional aging of 50 days (in total 218 days). Furthermore, T7482-1 and T7482-5 showed significant chemical changes after further additional aging of 30 days (in total 248 days).

Further additional aging for 30 days, mounting up to a total of additional 80 days (in total 248 days) of aging, at 70 °C at less than 10% RH included the membranes denoted T7482-1 and T7482-5 to the group of membranes that show significant chemical changes.

### 3.1.2. A-D Indicator Strips

The results suggested that none of the samples released acid, either before or after aging.

### 3.1.3. Beilstein Test

The results suggested that none of the samples contained PVC.

### 3.1.4. XRF

No inorganic elements were detected in samples of pure PE. By contrast, the Bruker software suggested that all membranes comprising regenerated PE contained calcium, titanium, and zinc by matching their spectral lines with internal references.

### 3.1.5. SEM

Examples of surface examination by SEM are shown in Figure 14 (unaged samples) and Figure 15 (aged). The area examined was approximately 600 μm long.

The SEM images were chosen as representatives of pure and regenerated PE respectively. The chosen SEM images show the differences, where T7482-4 (100% PE) appears to be homogenous in the surface structure both before and after aging, whilst many inclusions are apparent in T7482-9 (regenerated PE).

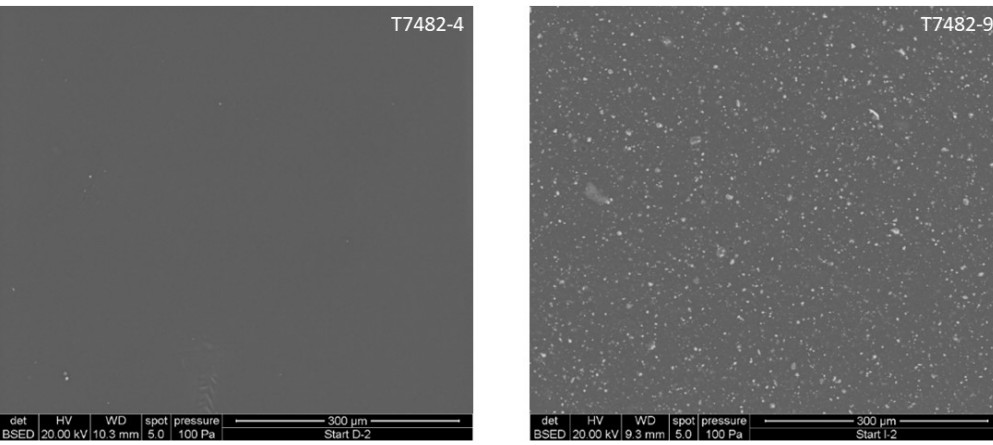

**Figure 14.** SEM of unaged membranes T7482-4 (**left**) and T7482-9 (**right**).

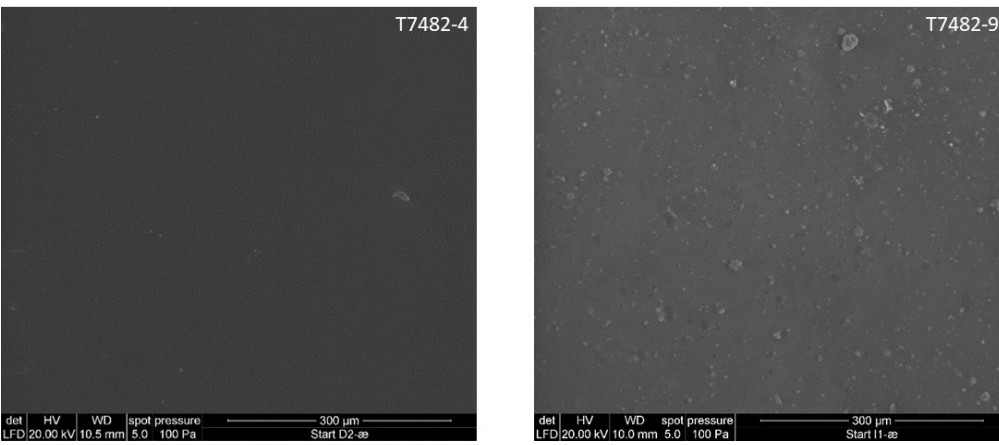

**Figure 15.** SEM of aged membranes T7482-4 (**left**) and T7482-9 (**right**).

### 3.1.6. Loss on Ignition

Loss on ignition for the samples examined are shown in Figure 16. The mean of two repeat measurements for each membrane are denoted as sample A and sample B. The mean loss on ignition was 99.94%, 99.98%, 99.82%, 100%, 98.91%, 98.29%, 99.63%, 98.89% and 88.47% for T7482-1, T7482-2, T7482-3, T7482-4, T7482-5, T7482-6, T7482-7, T7482-8 and T7482-9, respectively.

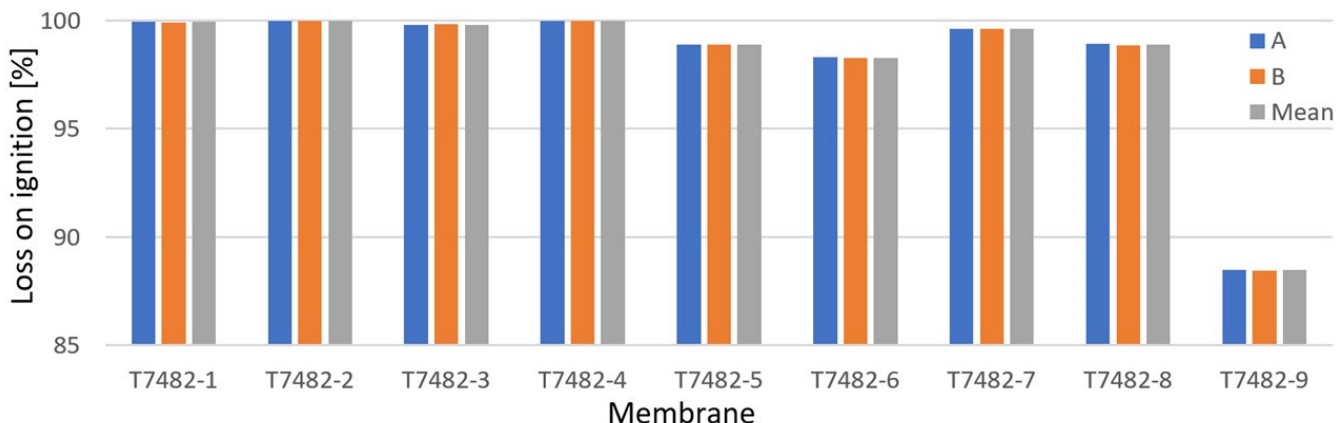

**Figure 16.** Loss on ignition for the samples examined.

Loss on ignition quantify the amount of material that anneals at a temperature of 550 °C. Loss of ignition was quantified as weight loss on annealing and calculated as weight loss divided by the weight of the sample prior to annealing, expressed in percent. The weight of the sample after annealing belongs to whereas inorganic additives and mineral impurities that will amount by higher temperatures. A loss on ignition of 100% quantifies that the sample was fully annealed.

### 3.2. Physicomechanical Tests

3.2.1. Tensile Strength

Results of tensile strength of membranes parallel to the direction of production, fracture elongation and Young's modulus performed on unaged and aged samples are shown in Table 4. Results are given as the mean value.

**Table 4.** Properties parallel to the direction of production.

| | Before Aging | | | After Aging | | |
|---|---|---|---|---|---|---|
| ID Number | Tensile Strength [MPa] | Fracture Elongation [%] | Young's Modulus [MPa] | Tensile Strength [MPa] | Fracture Elongation [%] | Young's Modulus [MPa] |
| T7482-1 | 17.5 | 456 | 105 | 16.0 | 402 | 133 |
| T7482-2 | 29.0 | 593 | 271 | 24.5 | 531 | 210 |
| T7482-3 | 20.7 | 402 | 133 | 19.2 | 350 | 153 |
| T7482-4 | 16.7 | 522 | 177 | 15.4 | 485 | 119 |
| T7482-5 | 16.4 | 601 | 116 | 16.6 | 687 | 106 |
| T7482-6 | 15.5 | 480 | 158 | 14.9 | 504 | 176 |
| T7482-7 | 18.9 | 623 | 162 | 18.0 | 646 | 153 |
| T7482-8 | 19.4 | 530 | 174 | 17.4 | 518 | 165 |
| T7482-9 | 14.9 | 556 | 158 | 14.0 | 571 | 191 |

Results of tensile strength along the direction perpendicular to the production direction, fracture elongation and Young's modulus performed on unaged and aged samples are shown in Table 5. Results are given as the mean value.

**Table 5.** Properties along the direction perpendicular to the production direction.

| | Prior to Aging | | | After Aging | | |
|---|---|---|---|---|---|---|
| ID Number | Tensile Strength [MPa] | Fracture Elongation [%] | Young's Modulus [MPa] | Tensile Strength [MPa] | Fracture Elongation [%] | Young's Modulus [MPa] |
| T7482-1 | 16.3 | 688 | 115 | 15.1 | 604 | 139 |
| T7482-2 | 28.4 | 747 | 274 | 26.6 | 688 | 245 |
| T7482-3 | 20.3 | 609 | 139 | 18.9 | 576 | 133 |
| T7482-4 | 16.2 | 652 | 89 | 15.3 | 623 | 109 |
| T7482-5 | 16.4 | 691 | 156 | 16.4 | 718 | 162 |
| T7482-6 | 18.7 | 641 | 168 | 18.0 | 649 | 165 |
| T7482-7 | 18.8 | 688 | 187 | 18.7 | 769 | 142 |
| T7482-8 | 18.4 | 710 | 203 | 16.6 | 670 | 172 |
| T7482-9 | 15.1 | 621 | 126 | 16.3 | 664 | 180 |

3.2.2. Tear Strength

Tear strength measured both parallel and along the direction perpendicular to the production direction on unaged and aged samples are shown in Table 6. The measurements are rounded to nearest 5 N, cf. standard [22]. Results are given as the mean value.

**Table 6.** Tear strength.

| | Prior to Aging | | After Aging | |
|---|---|---|---|---|
| **ID Number** | **Parallel [N]** | **Right Angles [N]** | **Parallel [N]** | **Right Angles [N]** |
| T7482-1 | 55 | 60 | 60 | 60 |
| T7482-2 | 85 | 85 | 90 | 90 |
| T7482-3 | 80 | 80 | 90 | 90 |
| T7482-4 | 60 | 55 | 60 | 60 |
| T7482-5 | 85 | 80 | 90 | 90 |
| T7482-6 | 115 | 115 | 115 | 115 |
| T7482-7 | 115 | 115 | 130 | 130 |
| T7482-8 | 110 | 110 | 115 | 115 |
| T7482-9 | 75 | 75 | 85 | 85 |

### 3.2.3. Resistance to Impact

Resistance to impact for unaged and aged samples is shown in Figure 17. Resistance to impact quantifies the impact at which the membrane was perforated. The resistance to impact was expressed as the maximum height from which a weighted ball can be released without penetrating the specimen. The drop weight falls from up to 2 m above the specimen. Resistance to impact of 2 m, quantify that the weighted ball can be released without penetrating the specimen. Results are given as the mean value.

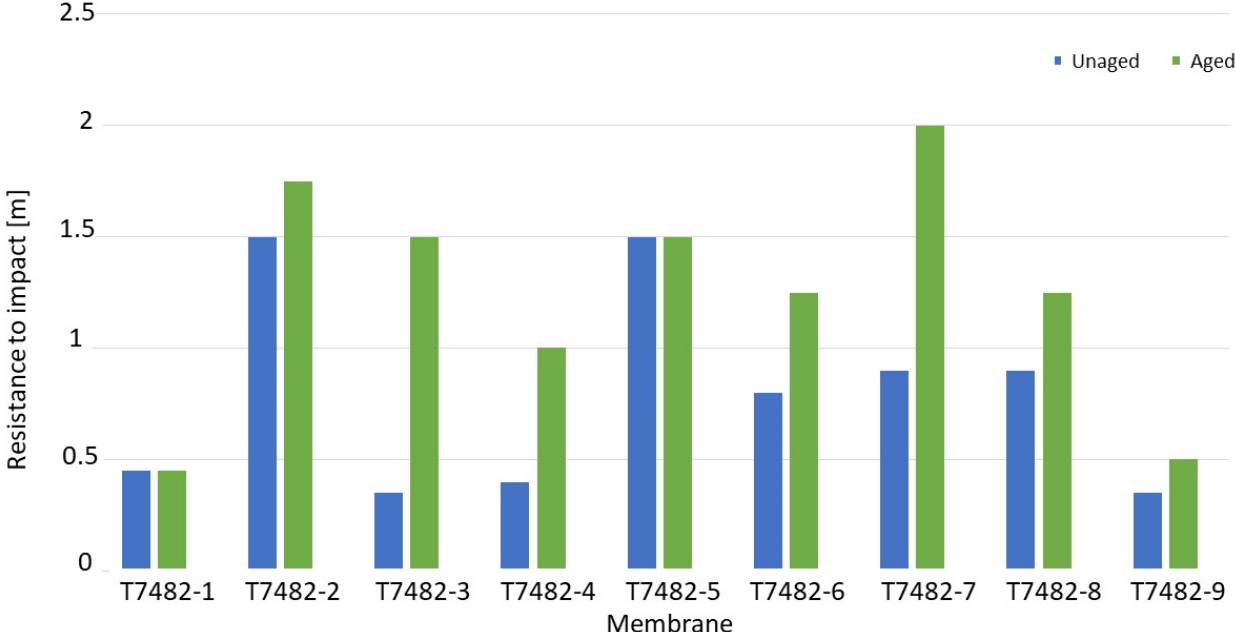

**Figure 17.** Resistance to impact for unaged and aged samples of membranes.

### 3.2.4. Mass and Thickness

Mass per unit area and thickness is shown in Table 7. Results are given as a mean value.

**Table 7.** Thickness and mass before aging.

| | Before Aging | |
|---|---|---|
| ID Number | Mass [g/m$^2$] | Thickness [mm] |
| T7482-1 | 141 | 0.161 |
| T7482-2 | 109 | 0.147 |
| T7482-3 | 182 | 0.206 |
| T7482-4 | 153 | 0.169 |
| T7482-5 | 123 | 0.184 |
| T7482-6 | 163 | 0.225 |
| T7482-7 | 171 | 0.220 |
| T7482-8 | 152 | 0.167 |
| T7482-9 | 105 | 0.156 |

3.2.5. Water Vapor Diffusion Resistance

Water vapor diffusion resistance determined on unaged and aged samples are shown in Figure 18. Water vapor diffusion resistance quantifies the diffusion of water vapor through the specimen. The water vapor diffusion resistance was quantified by recording the weight change over time in order to determine a constant moisture flow.

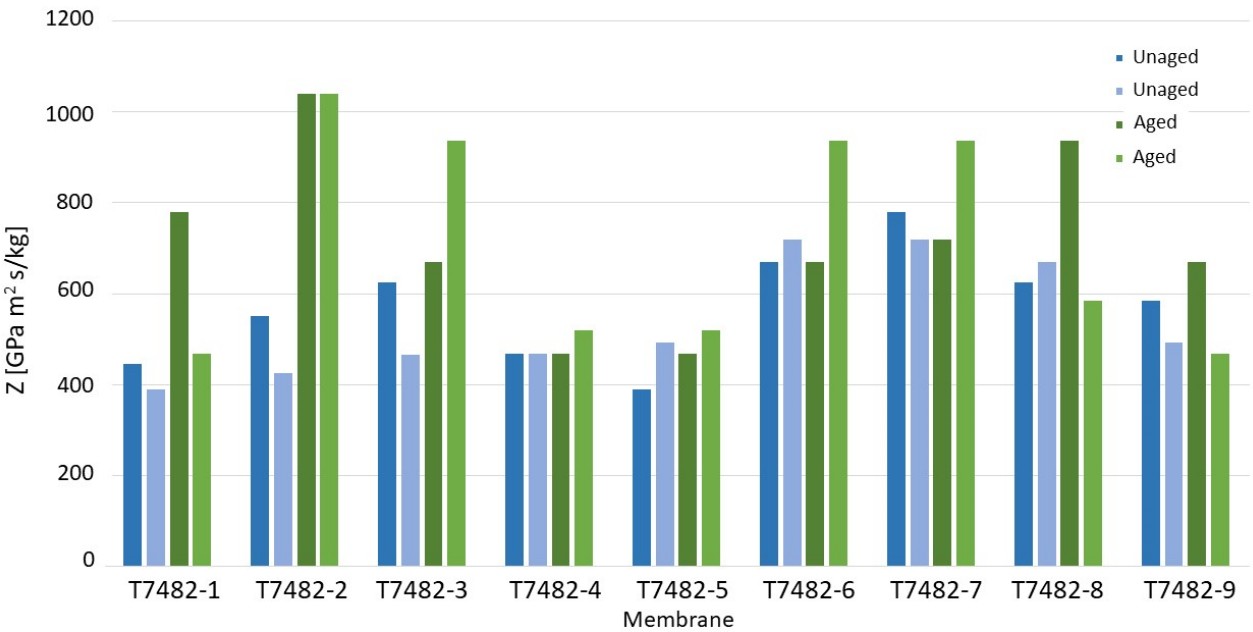

**Figure 18.** Water vapor diffusion resistance for the samples examined.

3.2.6. Surface Tension

Surface tension is shown in Figure 19. Dyne level of the surface tension for front side (with print) and backside of the unaged and aged membranes. Surface tension was quantified using liquids providing different contact angles of droplets on the membrane. Surface tension was quantified by a "duvet level", which can be used to compare the surface tension of membranes. Results are given as the mean value.

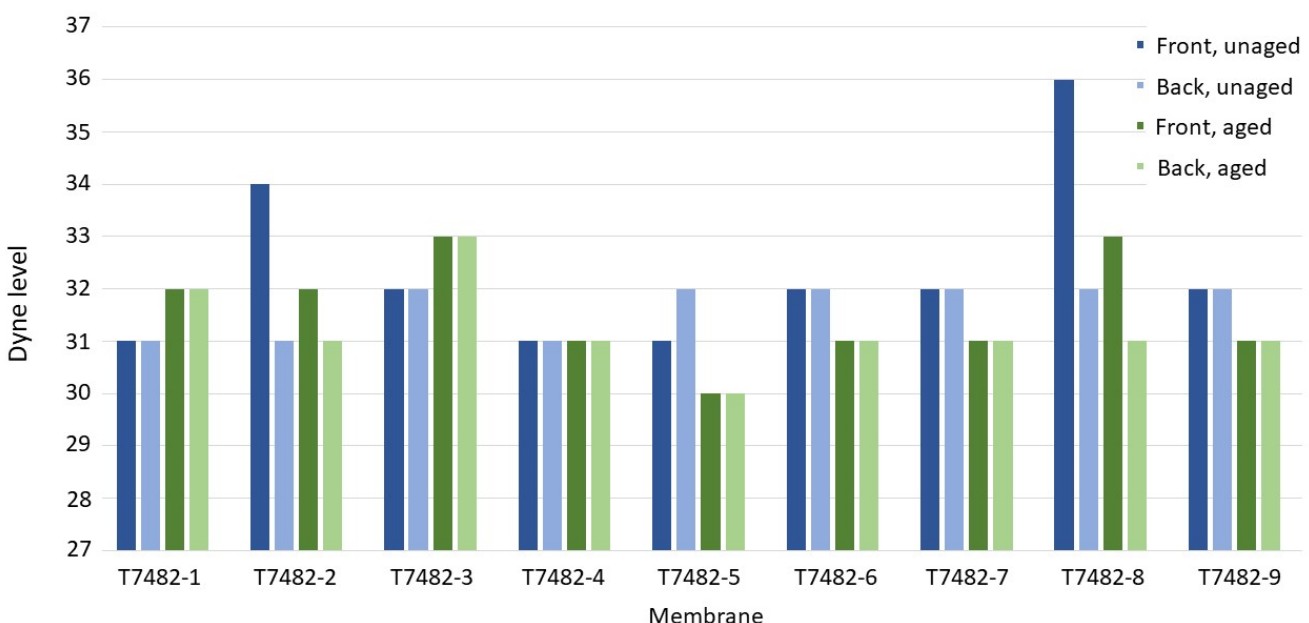

**Figure 19.** Surface tension expressed as the dyne level for the front cover (with print) and backside of the unaged and aged samples examined.

### 3.2.7. Adhesive Strength

Adhesive strengths of samples assembled with tape and butyl adhesive for unaged and aged samples are shown in Table 8. Results are given as the mean value of the highest strength achieved. The mean value is the average of seven samples for each vapour barrier system, expressed by a mean value and scatter.

**Table 8.** Adhesive strength [N/50 mm].

| | Before Aging | | After Aging | |
|---|---|---|---|---|
| **ID Number** | **Tape** | **Butyl** | **Tape** | **Butyl** |
| T7482-1 | 33.0 ± 3.7 | 47.7 ± 3.5 | 0.4 ± 0.1 | 56.0 ± 3.5 |
| T7482-2 | 53.9 ± 1.7 | 57.9 ± 2.2 | 42.7 ± 2.8 | 60.8 ± 2.8 |
| T7482-3 | 43.8 ± 1.8 | 58.5 ± 1.9 | 19.6 ± 5.2 | 58.0 ± 3.0 |
| T7482-4 | 62.8 ± 1.7 | 51.2 ± 0.8 | 25.1 ± 5.0 | 53.9 ± 3.7 |
| T7482-5 | 46.3 ± 1.0 | 54.0 ± 2.5 | 2.6 ± 0.2 | 55.8 ± 3.2 |
| T7482-6 | 46.4 ± 3.2 | 57.2 ± 4.2 | 1.9 ± 0.2 | 59.0 ± 2.0 |
| T7482-7 | 66.8 ± 1.3 | 52.9 ± 1.7 | 10.10 ± 1.3 | 61.7 ± 2.3 |
| T7482-8 | 56.5 ± 6.6 | 49.3 ± 4.0 | * | 59.9 ± 2.0 |
| T7482-9 | 35.0 ± 6.5 | 49.3 ± 2.0 | 4.6 ± 1.6 | 50.9 ± 0.6 |

* All seven specimens of taped joints of the vapour barrier system T7482-8 broke after aging by insertion into the jaws of the Instron and therefore could not be measured.

### 3.2.8. Bending Force

Bending force is shown in Table 9. As specimen T7482-1 has the lowest bending force it was assigned an index of 100. Membranes are for comparison indexed relative to membrane T7482-1. Results are given as the mean value.

**Table 9.** Bending force.

| ID Number | Young's Modulus | Thickness [m] | F [N] | Index |
|---|---|---|---|---|
| T7482-1 | 105 | 0.000161 | 1.75 | 100 |
| T7482-2 | 271 | 0.000147 | 3.44 | 196 |
| T7482-3 | 133 | 0.000206 | 4.65 | 265 |
| T7482-4 | 177 | 0.000169 | 3.42 | 195 |
| T7482-5 | 116 | 0.000184 | 2.89 | 165 |
| T7482-6 | 156 | 0.000225 | 7.20 | 411 |
| T7482-7 | 162 | 0.000220 | 6.90 | 394 |
| T7482-8 | 174 | 0.000197 | 5.32 | 304 |
| T7482-9 | 158 | 0.000156 | 2.40 | 137 |

*3.3. Visual Examination*

Number of holes and cracks along the fold-line for the individual membranes of PE are shown in Table 10.

**Table 10.** Examination of PE membranes.

| ID Number | Holes | Cracks in Fold | Fragments | Length [m] | Area [m$^2$] |
|---|---|---|---|---|---|
| T7482-5 | 0 | 0 | Yes | 125 | 250 |
| T7482-6 | 3 | 2 | Yes | 250 | 500 |
| T7482-7 | 0 | 0 | Yes | 250 | 500 |
| T7482-8 | 15 | 0 | Yes | 250 | 500 |
| T7482-9 | 0 | 0 | Yes | 250 | 500 |

**4. Discussion**

ATR-FTIR spectroscopy suggested that all membranes, both those comprising pure and regenerated PE, were based on LDPE. No other polymers were found using ATR-FTIR. The Beilstein Test proved negative for the presence of PVC. Signs of chemical degradation of PE on aging were detected by the development of carbonyl groups (C = O) in ATR-FTIR spectra attributed to oxidation of the plastic.

Development of carbonyl groups were only detected after the additional period of thermal aging for 80 days in total (248 days of aging). Based on chemical changes after aging, expected service lifetime of membranes can be divided into short, medium and long, as follows:

T7482-8 is the membrane is expected to have the shortest service lifetime
T7482-1 and T7482-5 are expected to have medium service lifetimes
T7482-2, T7482-3, T7482-4, T7482-6, T7482-7 and T7482-9 are expected to have the longest service lifetimes.

From the latter group, membranes T7482-3, T7482-6 and T7482-9 showed minor signs of oxidation whereas membranes T7482-2, T7482-4 and T7482-7 showed no evidence of oxidation.

The effects of oxidation of membranes were also reflected in the adhesive strength of taped joints for the vapour barrier system T7482-8. Aged samples of T7482-8 in contact with and in the immediate vicinity of the tape were brittle and the adhesive joints of all samples failed when inserted into the jaws of the Instron test machine (Figure 20).

The ability of membranes to produce acidic degradation products with time was negated by the results of A-D test. None of the membranes tested caused a change in colour of A-D strips from blue to green or yellow.

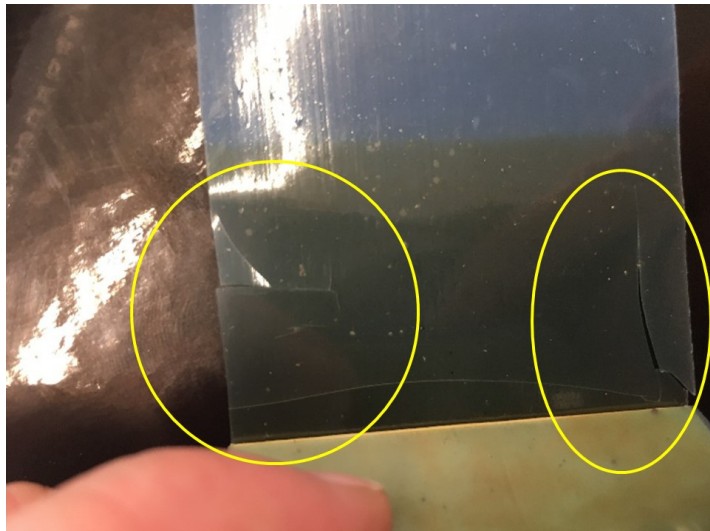

**Figure 20.** Membrane T7482-8 broke after aging for 168 days and could not be tested for adhesive strength when taped.

In addition to PE polymer, XRF detected elements calcium, titanium, and zinc in the form of calcium carbonate, titanium oxide and zinc oxide in all samples containing regenerated PE. It seems likely that these elements belong to the fillers added to many regenerated polymers to increase their strength and reduce opacity. Titanium oxide is also used to impart white colouration and to reflect UV light from the surfaces of the material. Together with zinc oxide, titanium oxide protects the polymer from sunlight and degradation. These inorganic additives were also detected using SEM. Samples of unaged and aged membranes showed a clear difference in the structure of the surface of the membranes, where membrane T7482-9 showed many inclusions compared with, for example, membrane T7482-4, which appeared to be more homogeneous in structure.

The quantity of inorganic additives in each membrane was determined using loss on ignition measurements. All membranes with virgin and 100% new PE had lower content of inorganic material (highest loss on ignition) compared to the membranes with regenerated PE. Only membrane T7482-4 comprised 100% organic and no inorganic material. Membrane T7482-9 had a loss on ignition of approximately 88.5% and thereby was the membrane containing the highest quantity of inorganic material.

Indexed flexural force of the investigated PE membranes sorted by easiest to work with are T7482-1, T7482-9, T7482-5, T7482-4, T7482-2, T7482-3, T7482-8, T7482-7 and T7482-6, respectively. It is seen that PE membrane T7482-6 has a bending force index which is four times higher than PE membrane T7482-1, which is due to a combination of its higher Young's modulus and greater thickness.

Tear strength showed no significant change for membranes on aging. However, it should be noted that the average tear strength of aged membranes was slightly higher than for unaged. The resistance to impact varied between the unaged membranes between 2.0 m and 0.45 m. Two of the PE membranes (T7482-1 and T7482-5) have the same resistance to impact before and after aging whereas the remaining seven PE membranes lost resistance to impact after aging, from 1.50 m to 0.35 m. No correlation was detected between resistance to impact and membranes containing pure PE and regenerated PE.

Water vapor moisture resistance number (Z-value) for PE membranes with a thickness of 0.15 mm is published as between 300 and 600 GPa m$^2$ s/kg [30]. In the current study, none of the membranes showed moisture resistances below 300 Gpa m$^2$ s/kg, and some showed above 1000 Gpa m$^2$ s/kg. All the membranes evaluated had satisfactory resistance to water vapor.

Surface tension tests on unaged samples showed the membranes T7482-2 and T7482-8 stood out by having more than one dyne level difference between the front and the back,

where the front has the highest dyne level. For aged samples, the difference in dyne level for the front and the back sides of these two membranes is smaller, but still such that the dyne level is highest for the front side. This means that when mounting membrane systems T7482-2 and T7482-8 one needs to be aware to tape joints at the front side of the membrane. However, the surface tension effect declines with time.

Tensile strengths for all the membranes were relatively similar at about 15–20 MPa. The single exception was membrane T7482-2 that showed significantly higher tensile strength, although the membrane was also the thinnest of those studied. This is accounted for when calculating the tensile strength in order to compare results. In general, variations in tensile strength between unaged and aged samples were small. Elongation at break along the direction perpendicular to the production direction for all membranes, both for unaged and aged samples, was between 550% and 750%, regardless of whether the PE membrane contains pure PE or regenerated PE. No effect of aging was detected based on Young's modulus. However, it should be noted that Young's modulus is subject to great uncertainty because it is calculated in a very narrow strain range.

Joints of PE membranes prepared with butyl adhesive were not weakened during aging. In contrast, the adhesive strength of vapour barrier systems prepared with tape was reduced on aging. The least effect of aging was found for the vapour barrier systems T7482-2, T7482-3 and T7482-4. For vapour barrier systems T7482-1, T7482-5 and T7482-6, the adhesive strength is below 3 N/50 mm for the aged samples. For all seven specimens of the vapour barrier system T7482-8, the membrane cracked in the immediate vicinity of the taped joint during handling, and it was therefore not possible to measure the adhesive strength. The adhesive strength was reduced after aging by respectively; 21% for membrane T7482-2, 55% for membrane T7482-3, 60% for membrane T7482-4, 85% for membrane T7482-7, 87% for membrane T7482-9, 94% for membrane T7482-5, 96% for membrane T7482-6 and 99% for membrane T7482-1 for taped joints.

Although T7482-8 joined using tape becomes very fragile after aging, this was not the case when it was joined using butyl adhesive. The fragility of the membrane was therefore assumed to be related to the behaviour of the tape or interaction between the tape and membrane during aging. Additionally, it was observed that the tape induced curling of the membrane, probably due to different coefficients of expansion during aging, caused by heat, moisture, or both, see Figure 21. None of the other tested taped joints exhibited similar behaviour.

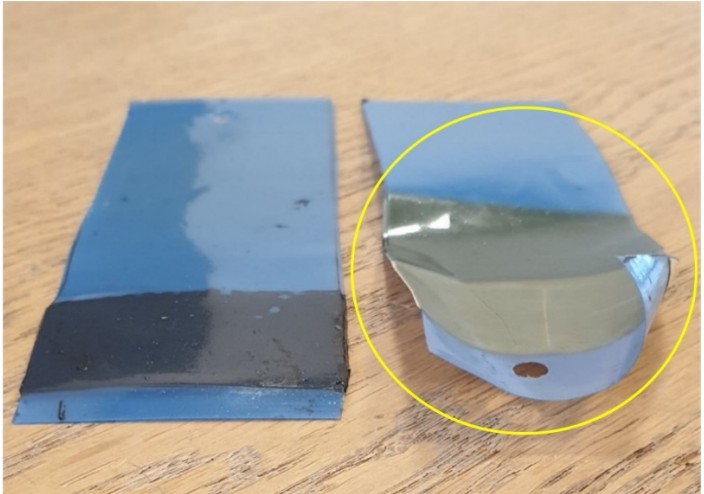

**Figure 21.** Samples of vapor barrier system T7482-8 after aging for 168 days, assembled with butyl (**left**) and assembled with tape (**right**).

Visual examination of membranes containing regenerated PE showed that all contained fragments of less than 1 mm and single fragments up to 4 mm. Cracks and holes

were found in membranes from several different rolls. For membrane T7482-6, three holes were found in the flat surface of the foil of the membrane and 2 holes in the membrane's fold-line. The holes were between 10 and 20 mm long and 3 mm wide. The holes in the membrane's fold-line were smaller than 3 mm. For membrane T7482-8, no holes were observed in the fold-line. By contrast, 15 holes were found in the flat surface of the membrane with dimensions up to 18 mm in length and 12 mm wide.

## 5. Conclusions

This study evaluated and compared the long-term performances of vapor barriers comprising virgin polyethylene (PE), 100% new PE, recycled PE and a combination of new and recycled PE. Membranes were evaluated both alone and as part of assembled systems. All membranes, tapes and adhesives were purchased for this study to retain independence.

The moisture barrier membranes and assembled systems purchased for this project were evaluated and compared using, amongst others, standard laboratory tests described by the standard [3], that is currently also accepted as a Danish standard. Evaluating tests were conducted on the materials before and after exposure to an aging regime comprising 168 days at 70 °C in total. The purpose of aging was to promote the chemical changes expected to take place within the normal service lifetimes of the vapour barrier systems so that they took place within a convenient period for the study. The aging regime used was equivalent to approximately 30 years of real time, calculated on the basis that the service temperature of the membranes and assembled systems is 10 °C (184 days aging at 70 °C). However, if the assumed usage temperature is increased to 20 °C, the real time corresponds to approximately 15 years. Materials were evaluated using six types of chemical analyses and 13 physicomechanical tests.

No general differences between the performance of membranes solely based on whether they were produced from pure PE (virgin PE and 100% new PE), regenerated PE or a combination of pure and regenerated PE were found. Importantly, water vapor diffusion resistance measurements showed that all membranes examined performed satisfactorily as vapor barriers when new and on aging. The PE membranes examined were found to be based on low density polyethylene, LDPE alone. All membranes examined were chemically stable and none of the membranes examined contained or produced acids upon aging.

Suppliers of the examined membranes suggest that regenerated PE comprises collected PE that is sorted, washed, purified, and stabilized with additives. The study shows that chemical and physical properties for regenerated PE membranes are similar to those of membranes produced from pure PE.

Visual examination of membranes containing regenerated PE showed that they all contain fragments smaller than 1 mm and up to 4 mm. For two out of five of the examined types of membranes of regenerated PE, fractures and holes were observed in the membrane between 3 and 20 mm. Based on visual examination, it may be concluded that a higher degree of quality is needed to produce vapor barrier membranes of regenerated PE. Producing membranes containing regenerated PE to be used for vapor barrier without significant defects is clearly a challenge for some manufacturers and is an issue that needs to be addressed.

Measurement of adhesive properties of taped membrane assemblies suggested that there was a general reduction in adhesion after aging, although one membrane appeared to interact with tape on aging and became so brittle that it fragmented before testing.

To examine their performance in the long term, additional aging of up to 80 days (in total 248 days) was performed on all membranes prior to re-examining them visually and chemically. No significant differences were measured in the performances of membranes comprising virgin PE, 100% new PE, recycled PE or a combination of new and recycled PE. However, it should be noted that chemical changes started taking place soon after the standard aging period described by the standard [13], which recommends aging according to [33], as used in this study. Based on the additional aging regime, the present study suggests the following:

T7482-8 is the membrane expected to have the shortest service lifetime;
T7482-1 and T7482-5 are expected to have medium service lifetimes;
T7482-2, T7482-3, T7482-4, T7482-6, T7482-7 and T7482-9 are expected to have the longest service lifetimes.

It was not possible to distinguish between aging properties for membranes comprising virgin PE, 100% new PE, regenerated PE or multi-layered virgin and regenerated PE.

**Author Contributions:** Conceptualization, methodology, resources and writing—original draft preparation, writing—review and editing, project administration and funding acquisition, T.V.R.; validation, formal analysis, investigation and data curation, Y.S., L.M.O., L.G.P., J.K.N., F.R.S., T.V.R. and T.K.H. All authors have read and agreed to the published version of the manuscript.

**Funding:** This research was funded by the National Building Foundation (Landsbyggefonden), the Danish Building Defects Fund (Byggeskadefonden), and the Landowners' Investment Foundation (Grundejernes Investeringsfond). Landsbyggefonden, Studiestræde 50, 1554 København V, CVR. NR: 6247 5412, EAN 5790001269265, E-mail lbf@lbf.dk, https://lbf.dk/om-lbf/english/. Byggeskadefonden, Studiestræde 50, 1554 København V, CVR: 12 02 59 71, E-mail: bsf@bsf.dk, https://bsf.dk/. Grundejernes Investeringsfond, Ny Kongensgade 15, 1472 København K, CVR nr: 26092515, EAN nr: 5798009811097, E-mail: gi@gi.dk.

**Institutional Review Board Statement:** Not applicable.

**Informed Consent Statement:** Not applicable.

**Data Availability Statement:** Data is reported in Danish: T.V.R., T.K.H., J.K.N., F.R.S., L.M.O., L.G.P., Hansen, M.H. and Y.S. 2020. Material Properties (In Danish: Materialeegenskaber—Test af polyethylenmembraners egenskaber før og efter accelereret ældning). SBi 2020:06. BUILD, Aalborg University. A literature study was performed before the preparation of the test program and is reported in Danish: T.V.R., Møller, E.B., F.R.S., J.K.N., L.M.O., L.G.P., Hansen, M.H. and Y.S. 2018. PE-membranes in the building envelope, a literature study (In Danish: PE-membraners levetid i byggeriet: et litteraturstudie). SBi-Report; No. 2018:11. Aalborg Universitet.

**Acknowledgments:** The authors thank The National Building Foundation (Landsbyggefonden), The Danish Building Defects Fund (Byggeskadefonden), and The Landowners' Investment Foundation (Grundejernes Investeringsfond) for financial support of this research project.

**Conflicts of Interest:** The authors declare no conflict of interest.

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
