# Peer review of "Performance of Polyethylene Vapor Barrier Systems in Temperate Climates"

_buildings, doi:10.3390/buildings12101768_

Round 1

Reviewer 1 Report

This manuscript assessed the performance of nine different vapor barrier systems comprising various PE. Six chemical analytical techniques and 13 physicomechanical tests were used to evaluate the samples’ performance. The author team confirmed that all the samples they evaluated performed satisfactorily from vapor barrier performance or mechanical performance. No significant difference in aging properties except for some PE samples (T8). Different adhesives, however, played a significant role during the aging process and they may deserve to study further in order to confirm the best way to keep the vapor barrier systems constant in years.

Overall, the research strategy is systemic and covers enough aspects to investigate the PE-based vapor barrier systems’ performance in temperate climates. However, the quality of the data, the method section, the result interpretation, etc. need to be further improved and modified. Some concerns/questions below arise from the reading of the paper, especially on the data analysis and method introduction. Please address them in the revised work for the publication of this manuscript in Buildings

Major concerns:

Data covers multi-sections (6 chemical analyses and 13 physicomechanical tests) but for each test, some missed the key information and some missed the key comparisons.

Information is not clear nor uniform. For example, how many aging days did the author team run the tests,  is it 168 days (Line 27) or 184 days (Line 154)?

Sample size, which is also referred to as the number of tested specimens, was not listed in the manuscript. And from the data shown in Table 4-7, they’re not like multiple specimens for one type of sample, thus, arising concerns about the data quality and validity.

Other concerns:

•           Renumber each section in the manuscript. The reviewer suggested combining your original sections 2 to 6 as a comprehensive section 2 to discuss Materials and Methods

•           Figure 1: Use arrows in the illustration, to show the membrane, overlapped area, joints, etc. PS: The grey part at the right bottom corner is a wall?

•           The reviewer strongly recommends the authors create a Table in Methods and Materials part to illustrate all the samples you tested and all 19 analyses or tests you used in this work.

•           Line 176: pure PE means 100% new PE here or virgin PE?

•           Line 202-206: The information of sample sizes (numbers) for each membrane used for different tests is suggested to list in Table 3. From your description, not all tests have the same size of samples.

•           Section 5.1: What are the dimensions of those dog-bone samples? Please list them

•           Section 5.2: 1) Lack of information about the mandrel/nail used to perforate the membrane; 2) please draft a figure to show the test apparatus.

•           Section 5.5: In the standard file DS/EN 1931, does it mention the conditions of RF1 and RF2? The reviewers just doubt that the RF values the authors used in this draft may not reflect the capability of WV resistance in the real life.

•           Figure 8: Mark the top view and front view for Fig 8A and 8B; and what’s the length of the adhesive area (shown as red areas, see attached snapshot)?

•     Section 5.8: there’s no bending stiffness symbol shown in equation 4. Please check it out and correct that. And also, do you have the proper apparatus for this bending test? For example, how do you monitor and record the deflection (d)?

•     Line 308-309, what’s your operating speed? 50 mm/min or 100mm/min?

•     Line 366: Except for T7482-9, I cannot see any obvious, additional peaks in the range of 1500-650 among T7482-5—T7482-8. Please either mark the additional peaks on the curves or modify your statement here.

•     Line 367-368: What’s the data to support your claim that “All samples containing regenerated PE contained calcium carbonate”? Please display the results to back up your statement.

•     Figure 12: Please include similar comparison curves for samples T3, T6, and T9. You can either put them here or in the supplementary file.

•     For Figures 12 and 13, unaged=0 days aging, aged=184 days aging? Additional aging=184+50=234 days aging? Please clarify the aging days either in the figure or in captions. It’s better to state them clearly in section 2 (methods).

•     Figure 13, do those PE membranes come as colorful film? How does your team remove the potential impact from colorful dyes?

·       Talking about oxidation, have you investigated or considered the impact from UV light? It is another important factor to oxidize polymer films, especially the ones used outdoors.

•     Section 7.1.4 XRF: where are the XRF spectrums for those membranes? Please add a new figure to show the metal elements from XRF spectrums

•     Section 7.1.5: What kind of message did the authors want to convey in this section? And why did you choose T4 and T9 as the representatives? And from Figure 12 and Figure 13, it seems like the aged groups (after 184 days) have no significant difference (either FTIR or visual observation) compared to additional gaining groups. It’s best to provide an SEM comparison between unaged and additional aging groups.

•     Line 420: “right angles to the direction of production”, do you mean to test the samples along the direction perpendicular to the production direction? If so, please change to this term; if not, please explain it with a figure or illustration.

•     Table 7: What’s the mass change after aging?

•     Figure 17, how many tests did the author team conduct for one type of membrane (unaged or aged)? From the plot, it seems only one test and it’s not valid.

•     Figure 18, why do you have 4 bars for one type of membrane? If you have two tests for each TYPE of Unaged/aged membrane, please use the “average ± std” and combine both in one bar.

•     Table 8: The strength has been recorded as the maximum force or the force at certain elongation? Please indicate that

•     Discussion Section

•     Line 473-476: Figure 12 only showed that T8 had s significant chemical changes after additional aging for 50 days, without any other spectrums (like T1 and T5), how did the authors tell that T1 and T5 are expected to have medium service lifetimes? And the rest membranes are expected to have the longest lifetimes?

•     Line 477-479, where were the minor signs of oxidation?

•     Line 490: Again, you need to display your XRF spectrums to show the elements contained in regenerated PE.

•     Line 530: Range is about 15-20 MPa? And also, you need more specimens to average those data

Author Response

Thank you for valuable review and suggestions to improve the manuscript. The proposals have been taken into consideration and the manuscript has been updated accordingly, as stated below for the individual concerns. However, the authors have been very concerned up on being precise in their use of wording and to distinguish between the use of the words: membrane, membrane system, and joint. Membrane is the foil; membrane system is assembled systems; and joint are tested individually.

Reviewer 2 Report

The manuscript did several tests on the performance of polyethylene membranes, and some valuable conclusions were drawn. I would recommend it for publication in buildings after minor revisions.

1.      The title seems confusing to me because the ‘temperature climates’ was not explained in the main text. Please consider revising the title or explaining the notion.

2.      Could more information be provided regarding the properties of membranes in Table1? For example, how the service life can differ so much between T7482-3 and T7482-4?

3.      It is suggested that the Abstract may be improved by mentioning the testing name instead of the details and concluding with more testing results.

4.      Keywords should be rewritten.

5.      Several typos: bolding in line 351, missing period in line 486 and 560.

Author Response

Thank you for valuable review and suggestions to improve the manuscript. The proposals have been taken into consideration and the manuscript has been updated accordingly, as stated below for the individual proposals.

Reviewer 3 Report

The paper deals with the long-term experimental analysis of nine vapor barrier systems available in the Danish market both from the chemical and the physico-mechanical point of view, considering also aging effects.  The document is relevant to the journal themes, it is fine organized and written in excellent English. In light of this, I recommend its publication after that the following minor aspects are addressed:

- the state-of-art is quite weak, please add some references with a similar topic also involving different materials for vapor barriers;

- in my opinion, some captions are too large(e.g. see Fig. 1 and Fig. 13), please short them;

- few lines of comments for every introduced graph after Fig. 16 and successive are appreciated;

Author Response

(The authors gave the same response as above.)

Round 2

Reviewer 1 Report

Thank you for your time to address the 1st round of comments. However, several major concerns and issues are found after reviewing your feedback.

1. The author should highlight your update, modified contexts, or images in the revised work and also indicate the location in the response report

2. Think about this question. What can we learn from this work? After so many tests and assessments, what did those data tell us or inspire the audience? What's the significance of this work? Did your data support your claims? (such as which group is best, and WHY it's best)

3. Some questions and concerns were not answered in this response report. The reviewer has listed them below and uses the RIGHT font to highlight them.

Please think more about the three major points mentioned above, and also address the following comments below from my 2nd round of reviewing.

Major concerns:

Data covers multi-sections (6 chemical analyses and 13 physicomechanical tests) but for each test, some missed the key information and some missed the key comparisons.

More details have been added to the descriptions og tests.

Highlight all of your new adding, revisions, or updates in the revised work, and also locate those changes in the new round response.

Information is not clear nor uniform. For example, how many aging days did the author team run the tests, is it 168 days (Line 27) or 184 days (Line 154)?

Thank you so much for the clearance, I am grateful. Ordinary aging is 168 days added by 50 days followed by 30 days at 70 °C.

Sample size, which is also referred to as the number of tested specimens, was not listed in the manuscript. And from the data shown in Table 4-7, they’re not like multiple specimens for one type of sample, thus, arising concerns about the data quality and validity.

Section 5 state: [Where’s your section 5?]

‘5. Physicomechanical Tests—[List your sample size in the table]

Five samples of each membrane (the water vapor diffusion resistance test was conducted on two samples of each membrane, and adhesive strength conducted on seven samples) were analyzed, using 13 physicomechanical tests. The tests were performed both on unaged and aged samples. Bending force (stiffness) was only conducted on unaged membranes. The tests performed, [11-18] are shown in Table 3.

and section 5.1 state:

‘Samples are shaped like meat bones, according to the specification described in [11], Type 5 in the specification.’

As an important addition to the results. it has been implemented in the paper, that test results are mean values.

Other concerns:

•           Section 5.1: What are the dimensions of those dog-bone samples? Please list them

‘Samples are shaped like meat bones, according to the specification described in [11], Type 5 in the specification.’—You should list them with dimension details in your work, even though you use the reference

•           Section 5.2: 1) Lack of information about the mandrel/nail used to perforate the membrane; 2) please draft a figure to show the test apparatus.

The fixture is shown in section 5.2 of DS/EN 12310-1:1999 Flexible sheets for waterproofing – Part 1: Bitumen sheets for roof waterproofing – Determination of resistance to tearing (nail shank) —I don’t think most of the audience have the access to this standard (it is sold on-line). Also, it’s the author team’s responsibility to make your method and statement clear. You’re suggested to draw an illustrated graph to describe your test apparatus.

•           Figure 8: Mark the top view and front view for Fig 8A and 8B; and what’s the length of the adhesive area ï¼ˆshown as red areas, see attached snapshot)?

Has been completed. —I haven’t seen the length of the adhesive area, no matter 8A or 8B. If the author team didn’t keep that length constant, the adhesive strength data may be doubtful.

•     Section 5.8: there’s no bending stiffness symbol shown in equation 4. Please check it out and correct that. And also, do you have the proper apparatus for this bending test? For example, how do you monitor and record the deflection (d)?

Bending stiffness has been changed to bending force in the article. I haven’t seen your answers to the rest of questions, like how do you monitor and record the deflection?

Section 7.1.5: What kind of message did the authors want to convey in this section? And why did you choose T4 and T9 as the representatives? And from Figure 12 and Figure 13, it seems like the aged groups (after 184 days) have no significant difference (either FTIR or visual observation) compared to additional gaining groups. It’s best to provide an SEM comparison between unaged and additional aging groups.

Correct; no significant difference was drawn from the SEM.—first, the author team didn’t answer the reviewer question, why did you choose T4 and T9 as the representative? 2nd, you have covered the details about T8 in Figure 12 and Figure 13, it’s reasonable to see the change from SEM images of T8 between unaged and additional aging groups. Unfortunately, the author team didn’t provide this comparison to confirm what you found from Figure 12 and Figure 13. Without any justification, using T4 and T9 seems like a random choice from your best groups. Please give a proper explanation, or replace SEM images of T4 with T8

•     Table 7: What’s the mass change after aging?

Has been corrected, Table 7. Thickness before and after aging, and mass before aging.—The reviewer wants to see the CHANGE of the mass. And if the change of mass means nothing, what’s the reason to list the mass before aging? You listed the thickness change here, what did you want to tell the audience? Same to the mass, if you just measure them and say you have such a data, that doesn’t make any sense in journal paper. It’s better to remember all the data you reported here should be useful or valid to draw the conclusion and inspire the other peers. Please re-edit this table and related context.

•     Discussion Section

•     Line 473-476: Figure 12 only showed that T8 had s significant chemical changes after additional aging for 50 days, without any other spectrums (like T1 and T5), how did the authors tell that T1 and T5 are expected to have medium service lifetimes? And the rest membranes are expected to have the longest lifetimes?

Aging was performed for 168 days at 70 °C. In addition, samples of membranes were exposed to an extended aging period comprising 80 days in a climate chamber at 70 °C and RH lower than 10 ±2% to further examine long term performance of membranes.—The author didn’t answer the questions. Do you have any data, to show that T1 and T5, has an inferior performance than the rest membranes? If so, please list them out and use them as the evidence to make your conclusion HERE. 

Author Response

Thank you for your time to address reflects on your 1st round of improvements of the article, hopefully additional clarifications will be satisfactory. I apologize for uploading the wrong file. In the newly uploaded file, all the changes according to the first review are marked. The changes from the 2nd review are marked as well.

Round 3

Reviewer 1 Report

It seems much better after 3rd round of revision and editing. Some minor concerns are still there and needed to be addressed. Please do that to make sure it has the best quality to publish in Buildings.

Minor concerns:

·       The revised work MUST come and display as the standard version, without any markup and tracked changes. Please confirm your revised work and hide all markup properly.

·       Remove Line 252 redundant image

·       Figure 8: Use the same scale for 25 mm in Fig 8(A) and Fig 8 (B)

·       Add some of your explanation in the result section. Like this paragraph “The SEM images were chosen as representatives of pure and regenerated PE respectively. The chosen SEM images show the differences, where T7482-4 (100% PE) appears to be homogenous in the surface structure both before and after aging, whilst many inclusions are apparent in T7482-9 (regenerated PE).” The readers will appreciate your detailed illustration.

·       Table 7: Considering you haven’t recorded the mass after aging, and also the thickness change has not been interpreted. Then it’s better to remove this part.

Author Response

Thank you for the suggestions to improve the paper, the minor concerns have been implemented. 

Minor concerns:

  • The revised work MUST come and display as the standard version, without any markup and tracked changes. Please confirm your revised work and hide all markup properly.
  • Remove Line 252 redundant image

The line 252 ‘The height from which the drop weight was released was determined by a split which was pulled out.’ has been deleted.

  • Figure 8: Use the same scale for 25 mm in Fig 8(A) and Fig 8 (B)

Figure 8 has been redrawn with same scale for 25 mm in Fig. 8 a) and Fig. 8 b).

  • Add some of your explanation in the result section. Like this paragraph “The SEM images were chosen as representatives of pure and regenerated PE respectively. The chosen SEM images show the differences, where T7482-4 (100% PE) appears to be homogenous in the surface structure both before and after aging, whilst many inclusions are apparent in T7482-9 (regenerated PE).” The readers will appreciate your detailed illustration.

The suggested paragraph has been added to the result section. Just below Fig. 15.

  • Table 7: Considering you haven’t recorded the mass after aging, and also the thickness change has not been interpreted. Then it’s better to remove this part.

Thickness after aging has been removed from the paper, in table 3 and table 7.